

# BioFuelDB: a database and prediction server of enzymes involved in biofuels production

Nikhil Chaudhary[*], Ankit Gupta[*], Sudheer Gupta and Vineet K. Sharma

Department of Biological Sciences, Indian Institute of Science Education and Research, Bhopal, Madhya Pradesh, India

[*] These authors contributed equally to this work.

## ABSTRACT

**Background**. In light of the rapid decrease in fossils fuel reserves and an increasing demand for energy, novel methods are required to explore alternative biofuel production processes to alleviate these pressures. A wide variety of molecules which can either be used as biofuels or as biofuel precursors are produced using microbial enzymes. However, the common challenges in the industrial implementation of enzyme catalysis for biofuel production are the unavailability of a comprehensive biofuel enzyme resource, low efficiency of known enzymes, and limited availability of enzymes which can function under extreme conditions in the industrial processes.

**Methods**. We have developed a comprehensive database of known enzymes with proven or potential applications in biofuel production through text mining of PubMed abstracts and other publicly available information. A total of 131 enzymes with a role in biofuel production were identified and classified into six enzyme classes and four broad application categories namely 'Alcohol production', 'Biodiesel production', 'Fuel Cell' and 'Alternate biofuels'. A prediction tool 'Benz' was developed to identify and classify novel homologues of the known biofuel enzyme sequences from sequenced genomes and metagenomes. 'Benz' employs a hybrid approach incorporating HMMER 3.0 and RAPSearch2 programs to provide high accuracy and high speed for prediction.

**Results**. Using the Benz tool, 153,754 novel homologues of biofuel enzymes were identified from 23 diverse metagenomic sources. The comprehensive data of curated biofuel enzymes, their novel homologs identified from diverse metagenomes, and the hybrid prediction tool Benz are presented as a web server which can be used for the prediction of biofuel enzymes from genomic and metagenomic datasets. The database and the Benz tool is publicly available at http://metabiosys.iiserb.ac.in/biofueldb & http://metagenomics.iiserb.ac.in/biofueldb.

## INTRODUCTION

The global increase in energy demand and decline in the available stock of fossil fuels has become a challenge and requires a search for alternate sources of fuels and energy. In this scenario, enzyme catalyzed conversion of biomass to biofuels provides an ideal source

Corresponding author
Vineet K. Sharma,
vineetks@iiserb.ac.in

of clean, ecological friendly and sustainable energy (*Fortman et al., 2008*). Bioalcohol and biodiesel are the two major commercial biofuels in use today. Bioalcohols including methanol, ethanol and higher alcohols such as butanol, can be produced through microbial fermentation of sugar (first generation biomass) or lignocellulosic biomass (second generation biomass) and can be directly used as fuel (*Canilha et al., 2012*). In contrast, biodiesel refers to structurally tailored fatty esters which can be used as a replacement for traditional petroleum fuels (*Steen et al., 2010*). Biodiesel can be produced from edible and non-edible oils such as animal fats, cooking oil, algae oil and microbial lipids, through a trans-esterification process (*Wang et al., 2015*). Although the demand for biodiesel is increasing, the use of biodiesel has a disadvantage of increasing the prices and reducing the availability of vegetable crops for meeting the global food demand. Hence, in this scenario, microbial fermentation of biomass (including non-food crops) for the production of biodiesel serves as an ideal alternative medium (*Oyola-Robles et al., 2014*). The common strategies involving microbial fermentation for biodiesel production include enhancing the expression of enzymes involved in the production of biodiesel precursor, such as fatty acids and triglycerides, and deletion of genes involved in the fatty acid degradation pathways (*Oyola-Robles et al., 2014*; *Tamano et al., 2013*).

The third major category of biofuels includes the microbial and enzymatic fuel cells. Microbial fuel cells are devices where microbes grow on the organic content and generate electric current. However, enzymatic fuel cells utilize cell free enzymes as electrodes for achieving the same functionality. In the previous studies, enzymatic fuel cells have been constructed using enzyme-mediated redox reactions at either or both of the electrodes (*Kakehi et al., 2007*; *Yuhashi et al., 2005*). In addition to these three biofuels, other alternatives such as terpenes, fatty aldehydes, fatty alcohols, biogas, etc., have been explored with varied success (*Kung et al., 2014*; *Lennen & Pfleger, 2013*; *Zieminski, Romanowska & Kowalska, 2012*).

Despite the merits of producing biofuels using enzymes, only a limited number of enzymes have been commercially exploited. It is primarily because of the unavailability of efficient enzymes that can perform the desired conversion and their inability to adapt to the application conditions which may not be optimum for a given enzyme, hence leading to a decrease in efficiency (*Bhardwaj Ajay, Zenone & Chen, 2015*). Consequently, the total share of all biofuels is 4% of global road transport fuel demand because of the higher cost of production compared to gasoline, low quality and uneven composition of the end products, low efficiency of biofuel enzymes, and loss of functionality of enzymes in adverse pH/temperature/chemical composition of the culture medium of the reactors (*Bhardwaj Ajay, Zenone & Chen, 2015*).

Therefore, to look for efficient and novel variants of enzymes involved in the different steps of biofuel production (referred to as 'biofuel enzymes' in the subsequent text), the first task is to expand our knowledgebase by exploring the naturally occurring biofuel enzymes and to search for homologues of these enzymes from natural environments. The application of high-throughput sequencing technologies has revealed the sequences of thousands of genomes which promise to facilitate the discovery and identification of novel biofuel enzymes. Furthermore, metagenomics has been developed into a culture

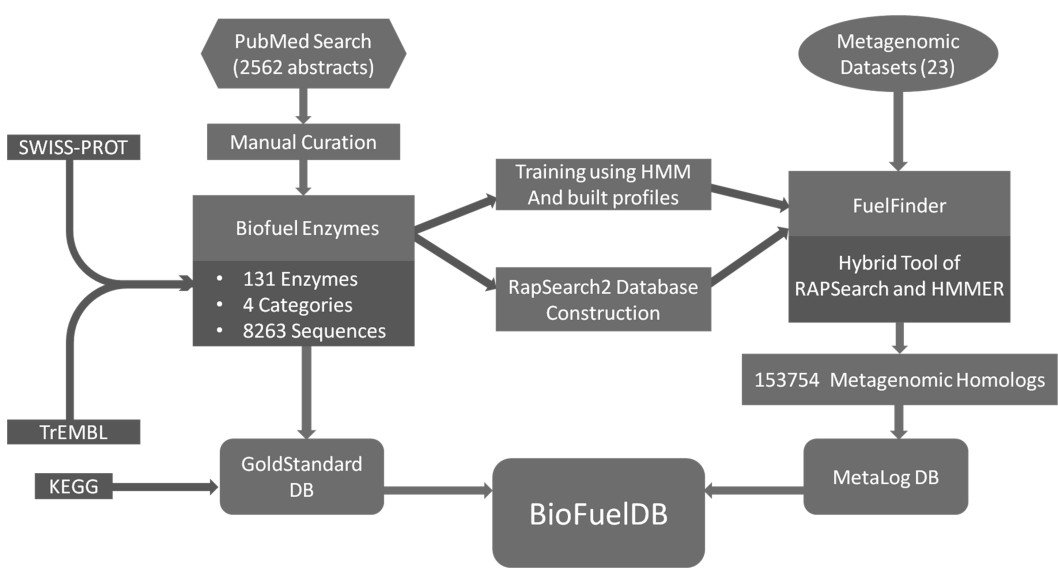

**Figure 1** Flowchart of the strategy used for constructing BioFuelDB.

independent approach that explores the diversity and complexity of microbial genomes in their natural environments and provides the information on novel genes and pathways from yet unculturable genomes. Thus, genome sequencing and metagenomics would assist in increasing the enzyme repertoire by revealing novel biofuel enzymes, and can also provide us with the functional variants of the existing enzymes. The availability of multiple metagenomic databases provides a useful opportunity to discover novel homologs of existing biofuel enzymes.

Several studies and databases have reported enzymes that can be used for biofuel production (*Choi et al., 2013*; *Lombard et al., 2014*; *Misra et al., 2016*; *Yin et al., 2012*). For example, the database of enzymes of microbial biofuel feedstock (dEMBF) provides information related to algal biofuel research but is limited to only 15 sequenced microalgal species and their implication in lipid synthesis (*Misra et al., 2016*). At present, there is no comprehensive database which is dedicated for retrieving information on biofuel enzymes. Hence, in this work, we have developed BioFuelDB, a comprehensive database of enzymes with demonstrated or potential application(s) in biofuel production by searching the available scientific literature. Using this database, a prediction tool 'Benz' was constructed by exploiting the homology-based and profile-based approaches to search for the potential homologs of existing biofuel enzymes. BiofuelDB is freely available at http://metabiosys.iiserb.ac.in/biofueldb and http://metagenomics.iiserb.ac.in/biofueldb.

## METHODS

### Construction of BioFuelDB

The flowchart of the methodology used for the construction of BioFuelDB is shown in Fig. 1.

### List of curated biofuel enzymes

The initial database of enzymes was constructed by searching the available 'English' abstracts containing the terms 'biofuel AND enzyme', 'biodiesel AND enzyme', 'alcohol AND enzyme', 'ethanol AND enzyme', 'methanol AND enzyme', 'fuel cell AND enzyme' and 'alternate biofuel' at NCBI PubMed and were imported to into a MySQL Database version 14.14 (*NCBI Resource Coordinators, 2013*). The initial set of candidate enzymes were identified using the 'Natural Language full-text search' and 'Boolean full-text search' features of MySQL. From this initial set of 2,562 abstracts, enzymes having demonstrated applications in the production of any category of biofuel(s) were manually curated. For the resultant 131 enzymes, the Enzyme Commission (EC) numbers were used as the unique identifiers to refer to individual enzymes in the BioFuelDB database. Based on their known application, the enzymes were classified into four broad categories namely 'Alcohol production', 'Biodiesel production', 'Fuel Cell' and 'Alternate biofuels'. An enzyme could be classified into more than one application category based on its known applications. All enzymes which are known to be involved in different steps of biofuel production are referred to as 'biofuel enzymes' in this manuscript.

### Enzyme sequence collection

Protein sequences for the curated list of enzymes were obtained from SwissProt database for all 131 enzymes. Sequences marked as 'putative', 'probable' or 'hypothetical' were removed from this initial set of sequences. The enzymes for which no SwissProt sequences remained after the removal of such sequences were discarded. SwissProt, which is a curated protein sequence database and provides high-quality annotation, was the preferred source for extracting the sequences over TrEMBL, which is a computer-annotated supplement to Swiss-Prot. However, for enzymes with less than five sequences available in the SwissProt database, sequences from TrEMBL were included. After following the above two steps, all enzymes with at least five representative sequences were included, and this database was termed as the Primary database. This primary database consisted of 8,263 sequences representing 131 selected enzymes.

### Other resources

Information about the reaction(s) catalyzed by the enzyme, its substrate(s), product(s), KEGG Orthology, and KEGG Pathways was obtained from the KEGG database (*Kanehisa et al., 2014*).

## Construction of 'Benz'

The Benz tool was developed to identify novel homologues of the known biofuel enzyme sequences from sequenced genomes and metagenomes. This tool employs a hybrid approach incorporating HMMER 3.0 and RAPSearch2 programs to provide high accuracy and high speed for prediction (*Eddy, 2008*; *Zhao, Tang & Ye, 2012*). It is a useful strategy to adopt a hybrid approach involving two different methods for predicting the novel homologs. The homology-based approach using RAPSearch2 would enable the direct detection of close homologs of the known biofuel enzymes. However, to overcome the limitations of homology-based pairwise alignment in detecting remote homologs,

profile-based methods implemented using HMMER 3.0 help to seek distant evolutionary relationships to detect remote homologs. Profile-based methods such as HMMER, consider the information of evolutionarily related sequences derived from multiple sequence alignments and gains sensitivity by including position-specific information into an alignment process by computing variation across related sequences at each position. Thus, HMMER3 predictions can be utilized by the user to detect more variant enzymes. To implement the HMM module, the protein sequences of the enzymes from the four categories were searched against the Pfam domains database. In total, 336 domains matched the sequences of the total set of 131 enzymes. However, several of these Pfam domains represent general functions (e.g., ATB binding domain, DNA binding domain, various loops etc.) and could not represent the specific function of the enzyme. Because of the lack of sufficient and conclusive information available through Pfam domains, HMM profiles were constructed for each enzyme. All sequences belonging to an enzyme were grouped together and aligned using the 'ClustalW' program (*Larkin et al., 2007*). The resulting alignment files were used as the input for 'hmmbuild' function of the HMMER 3.0 program to build HMM profiles. By combining HMM profiles of all 131 individual enzymes, a small HMM profile database was constructed exclusively for biofuels 'BioFuel-Pfam', which was further used as a reference database to identify homologs of the existing biofuel enzymes. As a parallel approach, the homology-based method 'RAPSearch2' along with profile-based 'HMMER3.0' was implemented to attain higher confidence in predictions using two different methods. The sequences representing individual enzymes as well as the different categories were processed using the 'prerapsearch' function of the RAPSearch2 program to build 'RAPSearch database' and were directly included in the Benz program. The two databases 'Biofuel-PfamDB' and 'RapsearchDB', constructed in the above sections were used as reference databases for the query sequences.

## Test datasets for the evaluation of Benz efficiency

Two test datasets were constructed to evaluate the performance of Benz program. In the first test dataset, a database of test sequences was prepared from the 'hypothetical', 'probable' and 'putative' sequences which were discarded during the preparation of the Primary database. This dataset consisted of 25,630 protein sequences. The results of the Benz output were compared with the known annotation of the sequences as well as with the results of the BLAST (blastp program with evalue cutoff 1e-6). To construct the second test dataset, ORFs from three different metagenomes (MG-RAST ids: mgm4466309, mgm4516289 and mgm4559623) were downloaded from the MG-RAST web server. A total of 549,870 ORF sequences from the three metagenomes were analyzed using Benz, and the results were compared using a BLAST search. The following standard parameters were used for accessing the efficiency of the program.

$$Sensitivity = \frac{tp}{tp+fn} \times 100$$

$$Specificity = \frac{tn}{tn+fp} \times 100$$

**Table 1** Distribution of biofuel enzymes in four application categories and six EC classes.

| Application category | Number of enzymes | | | | | | |
|---|---|---|---|---|---|---|---|
| | EC1 | EC2 | EC3 | EC4 | EC5 | EC6 | Total |
| Alcohol production | 20 | 8 | 33 | 10 | 2 | 1 | 74 |
| Biodiesel | 4 | 13 | 7 | 4 | 0 | 2 | 30 |
| Fuel cell | 19 | 0 | 3 | 2 | 1 | 2 | 27 |
| Alternate biofuels | 5 | 7 | 4 | 3 | 0 | 0 | 19 |

$$Accuracy = \frac{tp+tn}{tp+fn+tn+fp} \times 100$$

$$Matthews\ Correlation\ Coefficient\ (MCC) = \frac{(tp)(tn)-(fp)(fn)}{\sqrt{(tp+fp)(tp+fn)(tn+fp)(tn+fn)}}$$

where '$tp$' (true positives) are the sequences which belong to a given class of BioFuelDB enzymes and have been assigned to the same class by Benz, '$tn$' (true negatives) are the sequences which do not belong to a given class of BioFuelDB enzymes and have not been assigned, '$fp$' (false positives) are the sequences which do not belong to a given class but have been incorrectly assigned to it, and '$fn$' (false negatives) are the sequences which belong to a given class but have been incorrectly assigned to some other class. Only the hits with the tag 'C' were considered as acceptable results from Benz and were used for the performance assessment of the tool.

## Metagenomic datasets for mining biofuel enzymes

A total of 23 metagenomic datasets consisting of 22,470,288 ORFs were downloaded from MG-RAST and analyzed using "Benz" for the discovery of biofuel enzymes. These 23 selected metagenomes include sequences from diverse environments including marine, extreme saline, fresh water, aquatic, grasslands, hot springs, coral reef, extreme aquatic habitat (drilling), forests, village, activated sludge, hydrothermal vents, lakes, anthropogenic terrestrial biome and cropland.

# RESULTS AND DESCRIPTION OF THE WEB SERVER

## Distribution of enzymes across application categories

The distribution of 131 enzymes in four application-based categories provides an overall summary of the availability of enzymes for the production of various types of biofuels classified in the four categories. Out of the four categories, 'Alcohol production' contains the highest (74), followed by Biodiesel (30), Fuel Cell (27) and 'Alternate' which contains the lowest (19) number of enzymes (Table 1 and Table S1). This implies that almost half of the total numbers of known biofuel enzymes are involved in alcohol production. As mentioned earlier, selected enzymes may be present in more than one category if they were found to perform reactions involved in multiple categories.

Analysis of enzyme distribution across various EC classes in Primary database reveals some interesting observation. EC5 and EC6, representing 'Isomerases' and 'Ligases', contained least number of enzymes (three and four, respectively) in the database,

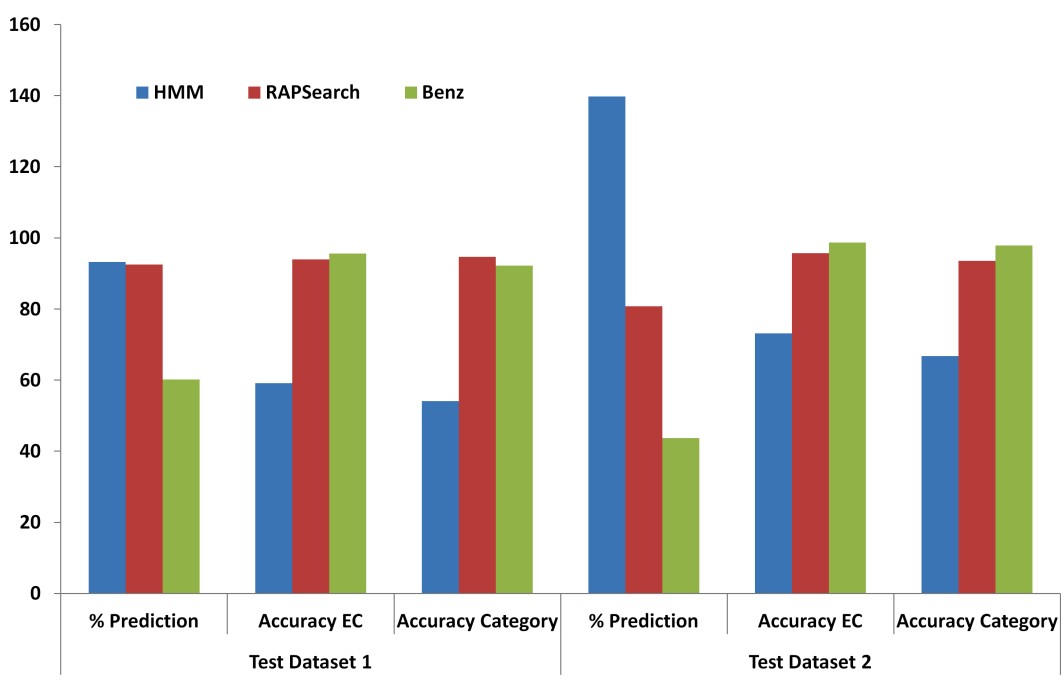

**Figure 2** Performance comparison of HMMER, RAPSearch and Benz tool in terms of percentage prediction, accuracy for EC Classes and accuracy for application categories.

whereas, the highest number of enzymes belonged to EC1 and EC3 classes, representing 'Oxidoreductases' and 'Hydrolases' (44 and 42, respectively) (Table 1 and Table S2). This can be ascribed to their high occurrences in various biological reactions, including those which lead to biofuel production. For 'Alcohol production' the largest percentage of enzymes in the Primary database was in EC1 and EC2 (50.44% and 20.99%), as these categories correspond to oxidoreductases and hydrolases, respectively. In case of 'Fuel Cells', majority of enzymes belonged to EC1 class, as oxidoreductases make good candidates for application at electrodes. In the case of 'Biodiesel production', the enzyme distribution was more widely spread across EC classes as biodiesel production occurs through a long chain of reactions employing a variety of enzymes.

## Performance evaluation of Benz

In case of the first test dataset, out of 24,009 sequences, Benz classified 23,014 sequences as biofuel enzymes, of which 14,456 were classified as 'consensus' results. From these consensus results, average accuracy values of 95.56% and 92.20% were observed for the enzyme classes and application categories, respectively (Fig. 2). RAPSearch2 was able to classify 22,196 sequences with an average accuracy of 93.89% and 94.68% for the enzyme classes and application categories, respectively (Fig. 2). Whereas, HMMER3 could classify 22,388 sequences with an average accuracy of 59.14% and 54.07% for various EC classes and application categories, respectively (Fig. 2). The other parameters of performance evaluation of test dataset 1 are shown in Tables S3 and S4.

In case of dataset 2, the predictions of Benz, RAPSearch and HMMER were compared with the results from BLAST search since the annotations of the sequences of this dataset were

unknown. Out of 549,870 sequences, BLAST classified 16,678 sequences as biofuel enzymes, whereas Benz classified 23,317 as biofuel enzymes. As the BLAST annotation was available only for 16,678 sequences, the performance measurement of Benz was performed only on these set of sequences, and the other predictions made by Benz could not be evaluated. Benz classified 7,292 sequences as 'consensus' sequences with the average accuracy of 98.64% and 97.89%, respectively, for various EC classes and application categories (Fig. 2). Using only RAPSearch2, 13,468 sequences could be classified with an average accuracy of 95.68% and 93.48%, respectively, for various EC classes and application categories (Fig. 2). Similarly, using only HMMER3, 21,177 sequences could be classified with an average accuracy of 73.09% and 66.79%, respectively, for various EC classes and application categories (Fig. 2). The other parameters of performance evaluation of test dataset 2 are shown in Tables S5 and S6.

As evident from the performance evaluation of Benz on test datasets 1 and 2, the consensus results are reliable with high sensitivity, specificity, accuracy, and MCC values, although this performance comes at the cost of percentage prediction. Rapsearch2 also provides a good performance while maintaining a high percentage prediction value. HMMER3, however, displays a variable performance on different datasets as well as across different application categories and EC classes. Thus, 'consensus' results can be considered more reliable, whereas HMMER3 predictions can be utilized by the user to detect more variant enzymes. Further, the availability of a profile-based search provides additional options to the users to search for biofuel enzymes using an alternative approach.

## Identification of novel homologues in metagenomic datasets

A total of 23 metagenomes from a variety of biomes were downloaded from the MG-RAST server (http://metagenomics.anl.gov/). Using the Benz program with stringent $E$-values of E-6 for RAPSearch2 and E-21 for HMMER, a total of 153,754 novel homologues could be identified from these metagenomes (Tables 2 and 3). The distribution of these enzymes predicted from different metagenomes showed a similar distribution pattern as observed in the global distribution. Across all categories, the highest (85,138) number of enzymes were present in the Bioalcohol category, followed by the Fuel Cells category (51,841). Biodiesel and Alternate Biofuels had 22,143 and 25,391 sequences, respectively, which were similar to the patterns observed earlier in the test dataset2 (Tables 2 and 3). Across EC classes, EC1 and EC2 were the most abundant (57,398 and 31,602 respectively), whereas EC5 contained the least number of sequences (3,917).

From all considered metegenomes, around 0.5%–1.5% of total ORFs were identified as Biofuel enzymes, which is a significant indication of their prevalence. Cropland, forest and other biomes (such as grassland and village biome) with degrading biomass were found to contain a higher proportion (1.8%) of enzymes under the 'Bioalcohol' category. This implies the presence of a large number of alcohol producing (fermentative) enzymes in the microbiome of these environments (*Limayem & Ricke, 2012*). Similarly, considerable proportions (0.31–1.81%) of biofuel enzymes were found in aquatic, marine and extreme environments. Furthermore, 21,553 homologues were identified from five extreme habitats consisting of hypersaline, drilling and hydrothermal vent and hot spring biomes (Table 2).
**Table 2  Distribution of novel homologs of biofuel enzymes (metalogs) in different application categories and metagenomes.**

| Metagenome ID | Metagenome source | Total ORFs | Alcohol | Biodiesel | Fuel cell | Others | Total biofuels |
|---|---|---|---|---|---|---|---|
| mgm4440324 | Marine biome | 36,701 | 88 | 22 | 54 | 23 | 153 |
| mgm4440329 | Hypersaline | 150,513 | 175 | 37 | 93 | 36 | 278 |
| mgm4441050 | Hypersaline | 3,517 | 18 | 6 | 23 | 3 | 44 |
| mgm4441102 | Hydrothermal vent | 368,502 | 2,316 | 742 | 2,070 | 1,118 | 5,231 |
| mgm4443684 | Freshwater | 388,210 | 1,594 | 497 | 1,430 | 568 | 3,394 |
| mgm4448052 | Aquatic biome | 414,473 | 1,995 | 654 | 1,674 | 920 | 4,401 |
| mgm4449252 | Grassland | 78,039 | 801 | 224 | 475 | 261 | 1,433 |
| mgm4460449 | Hot spring | 762,819 | 5,456 | 1,928 | 4,388 | 2,419 | 11,972 |
| mgm4466309 | Coral reef | 176,426 | 1,414 | 493 | 1,041 | 457 | 2,922 |
| mgm4467029 | Large lake | 376,200 | 2,029 | 810 | 1,155 | 1,836 | 5,012 |
| mgm4477803 | Lake | 5,383,950 | 9,814 | 2,911 | 7,907 | 2,706 | 19,965 |
| mgm4478241 | Extreme aquatic babitat (Drilling) | 222,722 | 2,702 | 417 | 1,597 | 736 | 4,028 |
| mgm4479942 | Village biome | 83,867 | 761 | 216 | 529 | 274 | 1,455 |
| mgm4487639 | Forest biome | 457,998 | 882 | 215 | 451 | 152 | 1,455 |
| mgm4494621 | Activated sludge | 5,054,731 | 14,444 | 4,727 | 12,546 | 3,063 | 28,882 |
| mgm4516289 | Aquatic biome | 253,233 | 1,490 | 526 | 1,010 | 674 | 3,085 |
| mgm4523306 | Anthropogenic terrestrial biome | 3,007 | 18 | 2 | 5 | 1 | 24 |
| mgm4527699 | Cropland biome | 755,188 | 7,616 | 1,160 | 2,573 | 2,149 | 11,336 |
| mgm4528623 | Cropland biome | 238,739 | 871 | 275 | 658 | 376 | 1,860 |
| mgm4537095 | Mediterranean forests, woodlands, shrub | 360,982 | 1,653 | 541 | 824 | 415 | 2,629 |
| mgm4559623 | Aquatic biome | 120,211 | 771 | 155 | 401 | 291 | 1,285 |
| mgm4571849 | Aquatic biome | 4,464,190 | 16,310 | 3,234 | 6,262 | 4,120 | 24,525 |
| mgm4571867 | Aquatic biome | 2,316,070 | 11,920 | 2,351 | 4,675 | 2,793 | 18,385 |
| **Total** | | **22,470,288** | **85,138** | **22,143** | **51,841** | **25,391** | **153,754** |

The use of enzymes which can survive at high temperatures (thermostable enzymes) enables the bioprocess temperatures higher than the distillation point of ethanol, in which case the gasified biofuels are collected from continuous bioprocess preventing end product inhibition (*Yeoman et al., 2010*). Similarly, the enzymes which can survive in hyper saline environments can catalyze the process of degrading cellulosic biomass for biofuel production (*Begemann et al., 2011*). These analyses suggest that the available metagenomic data from different environments provide possibilities of identifying novel homologs of biofuel enzymes which could be commercially exploited.

## Description of BioFuelDB Web Server

The Primary database consisting of 8,236 sequences of biofuel enzymes and 153,754 sequence homologs of biofuel enzyme (metalog), which were mined from the metagenomes, were combined to form the BioFuelDB database which was incorporated in the Web Server.

**Table 3  Distribution of novel homologs of biofuel enzymes (metalogs) in different EC Classes and metagenomes.**

| Metagenome ID | Metagenome source | EC 1 | EC 2 | EC 3 | EC 4 | EC 5 | EC 6 | Total |
|---|---|---|---|---|---|---|---|---|
| mgm4440324 | Marine biome | 52 | 52 | 7 | 20 | 2 | 20 | 153 |
| mgm4440329 | Hypersaline | 118 | 66 | 26 | 20 | 5 | 43 | 278 |
| mgm4441050 | Hypersaline | 25 | 5 | 1 | 5 | 0 | 8 | 44 |
| mgm4441102 | Hydrothermal vent | 2,047 | 1,179 | 281 | 866 | 114 | 744 | 5,231 |
| mgm4443684 | Freshwater | 1,271 | 577 | 383 | 485 | 75 | 603 | 3,394 |
| mgm4448052 | Aquatic biome | 1,542 | 831 | 726 | 544 | 40 | 718 | 4,401 |
| mgm4449252 | Grassland | 590 | 222 | 285 | 118 | 32 | 186 | 1,433 |
| mgm4460449 | Hot spring | 3,992 | 2,876 | 878 | 1,619 | 214 | 2,393 | 11,972 |
| mgm4466309 | Coral reef | 1,075 | 629 | 240 | 426 | 53 | 499 | 2,922 |
| mgm4467029 | Large lake | 1,569 | 928 | 1,373 | 636 | 61 | 445 | 5,012 |
| mgm4477803 | Lake | 8,874 | 3,901 | 1,479 | 2,595 | 401 | 2,715 | 19,965 |
| mgm4478241 | Extreme aquatic habitat (Drilling) | 1,732 | 849 | 195 | 372 | 10 | 870 | 4,028 |
| mgm4479942 | Village biome | 671 | 238 | 194 | 132 | 29 | 191 | 1,455 |
| mgm4487639 | Forest biome | 694 | 316 | 94 | 169 | 25 | 157 | 1,455 |
| mgm4494621 | Activated sludge | 12,812 | 5,781 | 1,027 | 4,268 | 236 | 4,758 | 28,882 |
| mgm4516289 | Aquatic biome | 1,096 | 585 | 492 | 404 | 62 | 446 | 3,085 |
| mgm4523306 | Anthropogenic terrestrial biome | 6 | 3 | 12 | 2 | 0 | 1 | 24 |
| mgm4527699 | Cropland biome | 3,322 | 1,869 | 3,383 | 1,437 | 475 | 850 | 11,336 |
| mgm4528623 | Cropland biome | 627 | 292 | 457 | 194 | 59 | 231 | 1,860 |
| mgm4537095 | Mediterranean forests, woodlands, shrub | 843 | 906 | 37 | 191 | 10 | 642 | 2,629 |
| mgm4559623 | Aquatic biome | 470 | 248 | 210 | 140 | 35 | 182 | 1,285 |
| mgm4571849 | Aquatic biome | 8,303 | 5,317 | 5,008 | 2,039 | 1,072 | 2,786 | 24,525 |
| mgm4571867 | Aquatic biome | 5,667 | 3,932 | 4,404 | 1,871 | 907 | 1,604 | 18,385 |
| **Total** | | **57,398** | **31,602** | **21,192** | **18,553** | **3,917** | **21,092** | **153,754** |

## Explore page

The 'Explore' page of the web server allows the user to search the enzymes from the BioFuelDB database by enzyme name, Enzyme Commission (EC) number, enzyme's systematic name or enzyme's KEGG Reaction ID(s). The user can also browse the database by application category of the enzymes, i.e., 'Alcohol Production', 'Biodiesel Production', 'Fuel Cell', 'Alternate Biofuels' or all the enzymes irrespective of the application category. Selecting an enzyme name takes the user to a new page where the various information about the enzyme are displayed such as, systematic name, EC number, common name(s) of the enzyme, application category, chemical reaction undertaken by the enzyme, KEGG reaction ID(s), substrates and products of the enzyme's reaction, the biological pathways in which the enzyme is involved and KEGG Orthology. Furthermore, the page provides the UniProt sequences of the enzyme in FASTA format as well as the PubMed references on which the application of the enzyme in the given application category was demonstrated.

## Prediction

The 'Prediction' page of the BioFuelDB web server is designed to identify novel homologues of the enzymes available in the BioFuelDB database. This page employs a hybrid tool 'Benz'

consisting of 'Biofuel-PfamDB' and 'RapsearchDB' databases at the backend for the prediction of novel homologs of biofuel enzymes which can be used for the production of biofuels. The user can either provide raw FASTA sequence(s) of proteins or ORFs in the input box or upload a query FASTA file through the 'Upload' interface. The query can be made either against all enzymes in the database, enzymes from any one application category, or against any one selected enzyme. For RAPSearch, the default e-value for the search is E-6, and for HMM it is E-21.

The output is reported in tab-separated format with nine columns namely: 'Query', 'HMM Hit' (matching hit from the HMMER profile), 'HMM *E*-value', 'HMM Score', 'RAPSearch2 Hit' (hit from the RAPSearch2 database), 'RAPSearch2 E-value', 'RAPSearch2 Score', 'Tags' (comparison between the outputs from both modules), and 'Category' (application categories of the predicted EC number). Tag 'C' implies that the results from both the modules are in consensus, 'N' implies that result from both the modules are not in consensus, 'H' implies that only the HMMER module provided a result for the particular query, and similarly, 'R' implies that only RAPSearch2 provided a result for the particular query. Queries with no hits from either program are not reported in the results file.

### MetaLog page

The 'MetaLog' page provides links to download the metagenomic homologues of the 131 biofuel enzymes (length > 50 amino acids) present in the Primary database enzymes predicted from 23 varied metagenomes using the Benz tool.

## DISCUSSION AND CONCLUSION

The main motivation for this work was the unavailability of any specialized database which provides comprehensive information on enzymes involved in different types of biofuel production. The mining of literature revealed that only a limited number of enzymes involved in biofuel production are currently known from a limited number of genomes. Therefore, as the first step, we constructed the 'BioFuelDB' knowledgebase of all enzymes involved in biofuel production from the available literature. However, the limited repertoire of these enzymes becomes a limitation while selecting the enzyme variants which can perform the desired reaction under the industrial conditions that may not be optimum for the given enzyme, hence leading to decrease in efficiency (*Bhardwaj Ajay, Zenone & Chen, 2015*). Thus, to explore efficient and novel variants of enzymes involved in different steps of biofuel production, we have developed the Benz tool which can identify novel homologues of the known biofuel enzyme sequences from sequenced genomes and metagenomes. The hybrid approach incorporating HMMER 3.0 and RAPSearch2 programs provides high accuracy and high speed for prediction of biofuel enzymes. Further, it appears to be a useful strategy to adopt a hybrid approach involving two different methods as the homology-based method RAPSearch2 enables the identification of close homologs of the known biofuel enzymes, whereas, the profile-based method HMMER 3.0 helps to identify the remote homologs which show low sequence identity.

In the present scenario, metagenomic data generated from different environments comprising of sequences from culturable and unculturable microbial genomes can be

mined to improve the repertoire of biofuel enzymes by revealing novel biofuel enzymes as well as the functional variants of the existing enzymes. In this study, the identification of 153,754 enzymes from 23 metagenomes indicates the possibility of finding such enzymes by exploiting the metagenomic data from several hundreds of metagenomes. Furthermore, the metagenomes are so rich in microbial diversity and functional genes that it is almost certain to identify the novel variant of a given enzyme (*Sharma et al., 2010*). Thus, the mining of novel homologues of biofuel enzymes from different environments using their metagenomic data enables the identification of novel variants which can work in wide range of conditions, and thus improves the enzyme repertoire.

To our knowledge, BioFuelDB is the first comprehensive dataset of biofuel enzymes. We anticipate that it would act as a comprehensive resource of biofuel enzymes and would assist researchers to explore novel variants of biofuel enzymes from different environments. However, the efficiency of the novel variants can only be ascertained through laboratory experiments, but the high quality of the initial primary database and stringent search criteria of Benz tool ensures that all the predicted enzyme sequences can be used as leads for experimental validations. The database and tools are available freely at the website http://metabiosys.iiserb.ac.in/biofueldb & http://metagenomics.iiserb.ac.in/biofueldb.

### Funding
This work was supported by the Department of Biotechnology (Project number BT/PR7/863/BID/7/435/2013). Ankit Gupta is a recipient of DST-INSPIRE Fellowship from the Department of Science of Technology, India. The funders had no role in study design, data collection and analysis, decision to publish, or preparation of the manuscript.

### Grant Disclosures
The following grant information was disclosed by the authors:
Department of Biotechnology: BT/PR7/863/BID/7/435/2013.
DST-INSPIRE Fellowship.

### Competing Interests
The authors declare there are no competing interests.

### Author Contributions
- Nikhil Chaudhary and Ankit Gupta conceived and designed the experiments, performed the experiments, analyzed the data, contributed reagents/materials/analysis tools, wrote the paper, prepared figures and/or tables, reviewed drafts of the paper.
- Sudheer Gupta contributed reagents/materials/analysis tools, the website was constructed by this author.
- Vineet K. Sharma conceived and designed the experiments, analyzed the data, wrote the paper, reviewed drafts of the paper.

## Data Availability

The data related to this work is available at http://metabiosys.iiserb.ac.in/biofueldb & http://metagenomics.iiserb.ac.in/biofueldb.

## Supplemental Information

Supplemental information for this article can be found online at http://dx.doi.org/10.7717/peerj.3497#supplemental-information.

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
