# Peer review of "BioFuelDB: a database and prediction server of enzymes involved in biofuels production"

_PeerJ, doi:10.7717/peerj.3497_

## Round 0.1 · original submission · Major Revisions

Dear authors,
Your manuscript has been reviewed by three experts in the field. They have made several suggestions to improve the manuscript. Please read them and make the indicated corrections as suggested.

Reviewer 1 ·

Basic reporting

Dear Authors,

1. The manuscript lacks clarity in many section. It needs to be rewritten to improve presentation of results.

2. Reference of biofuel database should be included. Authors should clearly state new results and how BiofuelDB database is different from other biofuel resouces such as "http://bioinfo.nist.gov/biofuels/" and "http://bbprof.immt.res.in/embf/".

4. The statement needs clarity "However, to overcome the limitations of homology-based pairwise alignment in detecting low percent identity homologs, profile-based methods implemented using HMMER 3.0 would help to seek distant evolutionary relationships for the detection of remote homologs". For example, for 'homology-based pairwise alignment', authors wants to say 'overcome limitations of homology searches in detecting homologs with low sequence identity'.

5. needs fixing of 'cl+ass' word in line 167.

6. This statement needs clarity "Furthermore, metagenomics has emerged as a culture independent approach that explores the 73 diversity and complexity of microbial genomes in their natural environments and provides the 74 information on novel genes and pathways from yet unculturable genomes."

7. In the 'Background' section of Abstract, authors mentions that "unavailability of a comprehensive biofuel enzyme resource, low efficiency of known enzymes, and unavailability of enzymes which can function under extreme conditions in the industrial processes". First, I think there are biofuel enzyme resources. Secondly, enzymes from thermophilic bacteria can work in extreme condition. I think authors wants to say that there are no known enzymes yet used at industrial scale.

Experimental design

Dear Authors,

1. Since HMMER can detect both remote and close homologs, I think there is no need to use both HMMER and Rap2Search program to identify homologs. Authors need to specify reason to use Rap2search, may be it is more accurate in prediction.

2. It is surprising to see very absymal HMMER performance to classify enzymes under difference EC's. Even for EC 5 and 6, HMMER accuracy is barely 0.45. May be authors can comment on this.

3. Consensus approach used by authors is not mentioned explicity in the 'methods' section.

4. Is there reason for taking 'e-21' cut off for HMMER searches? Is it to reduce no. of false positives prediction. Authors can used query coverage as a condition to reduce false positives.

5. Metalog hits needs pruning. Short reads can be eliminated from searches, to make database more effective. Because having a homolog of 50 residues has no practical application.

6. Authors have provided sufficient details of method.

Validity of the findings

Dear authors,

1. Authors have used PubMed to find use of enzymes into 4 categories of biofuels. This is very useful to scientists trying to find suitable enzymes for industrial application.

2. Authors have used word 'novel homologs'. Given that homologs can be identified easily by normal sequence searches, I think 'novel' should be used carefully and only for really new findings.

3. Data presented by authors is well curated, except Metalog, which consists of ORFs from short reads. Such small sequence ORFs are of no use.

·

Basic reporting

The article submitted by Nikhil Chaudhary et al, entitled BioFuelDB: a database and prediction server of enzymes involved in biofuels production describes the first comprehensible dataset of biofuels enzymes. The authors defined four categories: Alcohol production, Biodiesel production, Fuel cell and Alternate biofuels. The dataset is available and it can be checked in http://metabiosys.iiserb.ac.in/biofueldb.
The text is clear and unambiguous and the authors use a proper English in the manuscript.
The references used are enough to support the mentioned information, as well as the discussion. During the Introduction section, the authors provided an adequate background (in general). However, I suggest that the authors write an Introduction more coherent with the main goal of the manuscript.
The manuscript is well structured, and in my opinion is relevant for the readers interesting in Metagenomics and related approaches, as well as biotechnologist.

Experimental design

The authors inform the first database and prediction server of enzymes involved in biofuels production. I have checked the server and as the authors mentioned it is public and is available. The work performed is relevant and necessary. The methods were written in detail and were totally comprensible for my understanding.
But I have a doubt, why did the authors select only 23 metagenomes?! Please, can you provide me an explanation?

Validity of the findings

The database and prediction server will benefit to the readers because is the first database with metagenomic information related with enzymes involved in biofuels and alcohol production.In my opinion is totally necessary and it is very useful.
Please, can you analyze if Tables 1 to 5 are transferred as supporting materials? In my opinion they are not necessary in the main text.

Additional comments

Very nice work!

Reviewer 3 ·

Basic reporting

Throughout the text, the English language should be improved for accuracy and clarity to allow the audience to easily access the ideas of your manuscript. Specific sections to focus on clarifying includes lines 191-202, 220-232, 315-318, and 327-330. I suggest you have a colleague proof-read the manuscript. A non-exhaustive list of improvements to the language and grammar is attached to the end of this review.

Experimental design

Several of the methods require additional information for clarity such as the type and method of the BLAST analyses. I suggest detailing the steps taken to perform these analyses in an additional supplemental code file.

Validity of the findings

No comment.

Additional comments

This study presents a needed database and a sound curation. Please consider the following comments to improve the text:

Please clarify the meaning of the term “131 enzymes” on Lines 19, 108, 140, and throughout the text. Please clarify whether this was 131 sequences, named enzyme type, or other identifier. From Figure 1, this number appears to be assigned for the names of the enzymes, representative of 8263 sequences.

For the keyword search outlined in Lines 92-95, please clarify whether the terms combined as one string such as “biofuel enzyme” or expanded so that the abstract would contain “biofuel” AND “enzyme”. Additionally, were both search methods tested?

For the enzyme sequence collection identified in Lines 107-113, what was the rationale for limiting the search to SwissProt sequences and not expand the entire search to the entire TrEMBL library? You will gain many more sequences in the TrEMBL library at the cost of lower curation accuracy of the annotations. If this was the intention for obtaining the highest quality annotations for the downstream assignments, then please explain this point here to assist readers not familiar with the architecture of SwissProt and TrEMBL.

The BLAST analyses that were employed, such as on Line 154, are insufficiently described within the text to ascertain the resulting output and were not included in the supplied Perl script. What program was utilized (e.g., BLASTx, BLASTn)? What were the parameter settings? What were the cutoffs selected (e-value or bitscore) and why was that threshold selected? These points likely are best addressed in the Materials and Methods.

L204-L232 : Please consider reorganizing the presentation of the performance metrics to reveal the desired comparisons. For example, the reader may benefit by grouping the results by number of sequences, the accuracy, and MCC values rather than by method performed to directly compare the relationships in text.

L244-245 : Stringent evalues for the Benz program and RAPSearch2 are set at E-6 and E-21, respectively. Please clarify how these values were calculated for the two programs and how they are comparable. Additionally, please provide a justification for ascribing these as ‘stringent’. The reader currently has no information directly within the text to judge these statements.

L332-334 : The statement that novel variants are almost guaranteed to be discovered in a metagenome should be supported by an appropriate reference.

L334-335 : Please take care with this assertion that the enzyme will be automatically the most suited for the environmental condition because the organism is present and expressing that specific unit. Because evolution operates on the full organism in addition to the enzyme, the expressed enzyme may actually not be optimized for the conditions because of multiple scenarios including an upstream pathway is unavailable or production of the “optimal” enzyme is too costly. Therefore, this statement would be more accurate when the phrasing reflects that resident organisms of a particular environment undergo a particular set of environmental selections that increase their likelihood of expressing enzymes that are more suited to the environment.

Figure 1 : Please clarify why two different colors are used for the arrows.

Table 1 : Please clarify whether any enzymes were not assigned an EC number.

Table 2 : Please show an appropriate number of decimal places in the footer: at maximum, five.

Tables 2-5 would be visually more comprehensible for faster comparisons as a graphic such as a bar chart. Consider presenting these as barcharts.

Table 6 : Please be consistent with the capitalization of the metagenome source names.

Table 7 : Please include a metagenome source names column in this table as well.

Suggested Improvements to the Language:

Throughout the document, please consider replacing “due to” with “because of”
Throughout the document, please check that a space is present between the terminal word in a sentence and the citation
L10 – Consider strengthening the sentence from “there is a need to explore novel methods of biofuel production” to “novel methods explore alternative biofuel production processes to alleviate these pressures”
L41 – Please replace “In contrast” with “By contrast”
L43 – Please add a comma after “microbial lipids”; please replace “by” with “through a”
L44 – Please replace though with “although”; consider replacing “it” with an appropriate noun for clarity such as “the use of biodiesel”
L49 – Please clarify the “precursors, such as, fatty acids, triglycerides”. Was this phrase intended as “precursors, such as fatty acids and triglycerides,”?
L53 – Please replace “on the other hand” with “however”
L54-55 – Consider moving “in previous studies” to the beginning of the sentence
L56 – Please add a comma after “etc.”. Alternatively, consider restructuring the list to remove the use of “etc”
L59 – Please remove the comma after “primarily”
L61 – Please add an “a” in front of “decrease”
L62 – Consider using a synonym for “Resultantly”; consider removing “a meager” to strengthen the tone and objectivity of the sentence
L64 – Please add an “and” before “loss of functionality”
L65 – Please remove the “etc.” at the end of the sentence
L66 – Please remove “in order”
L69 – Please remove “the” before both “homologues” and “natural environments”
L70 – Please remove “genome”, the usage is redundant with the genome on L71
L72 – Consider replacing “has emerged as” with wording such as “been developed into” to reflect the past work undertaken to curate this approach
L75 – Please replace “be very helpful” with “assist”
L77 – Please replace “numerous” with “multiple” because the number of databases can be counted; additionally, please consider removing “unique” because this approach can be applied to a broad range of enzymes, not exclusive to biodiesel related enzymes
L78 – Please remove the “the” before “existing”
L80 – Please remove the extra space after “however,”; please replace “comprehensive database till date” with “current comprehensive database”
L82 – Please replace “proven” with “demonstrated”
L83 – Please move “available” before “scientific literature”
L85 – Please remove “the” before “existing”
L97 – Consider replacing “proven” with “demonstrated”
L98 – Please remove the “-” before “curated”
L101 – Throughout the document, please consider writing out all numbers below ten in word form rather than numeric; additionally, be consistent with the formatting of the single-quotes throughout the document
L102 – To strengthen the sentence, please consider removing “It is to be noted that”
L109 – Consider replacing “were left” with “remained”
L111 – Please capitalize the “t” in “TrEMBL”
L116 – Please add an “and” before “KEGG Pathways”
L119 – Consider replacing “it” with an appropriate noun for clarity such as “this tool”
L125 – Please remove “would”
L126 – Consider replacing “for the detection of remote” with “to detect remote”
L127 – Please replace “takes into account” with “consider”
L128 – Please pluralize “alignment” to “alignments”
L129 – Please replace “alignment process” with “an alignment process by”
L131 – Please remove “all”
L132-133 – Please consider restructuring this sentence to wording similar to “In total, 336 domains matched the sequences of the total set of 131 enzymes.”
L133: Consider replacing “belonged to” with “represent”
L134: Please add a comma after “e.g.”
L137 : Please add “the” before “ClustalW”
L139 : Please remove “from these alignments”
L147 : Please remove the colon
L148 : Please add a comma after sections
L149 : Please add “the” before “Evaluation”
L154 : Please remove the comma after “sequences”
L156 : Please add “the” before “MG-RAST”
L157 : Please add a comma after “Benz”
L158 : Consider changing “with the BLAST program” to “using a BLAST search”
L163 : Please correct the formatting of the “h” in “Matthews”
L166 : Please remove “the” before “Benz”
L167 : Please correct “cl+ass”
L168 and 169: Please remove “to it”
L175 : Please remove “website”
L176 : Consider replacing “, and include” with “including”
L184 : Please add “the” before “production”
L185 : Please consider removing “Interestingly,”
L186-187 : Please be consistent with presenting names of categories within single quotes
L187 : Please add “the” before “total”
L188 : Consider replacing “some” with “select”
L189 : Please remove the comma after “category”
L234 : Please replace “though” with “although”
L235 : Please replace “very” with “a”
L236 : Please replace “while” with “whereas”; replace “on the other hand” with “however”
L237 : Please consider replacing “gives” with “displays a”; replace “at” with “on”
L238 : Please remove the comma after “whereas”
L239 : Please check and correct the capitalization of “HMMer3” when needed
L240 : Please add “a” before “profile-based”
L247 : Please replace “case of” with “the”
L248 : Please add “the” in front of “Bioalcohol”
L249 : Please add “the” in front of “Fuel Cells”
L250 : Consider replacing “echoing” with “similar to”
L253 : Consider removing the sentence “The analysis of these metagenomic enzymes reveals some noteworthy findings” because this phrase does not provide additional information for the reader and does not substantially assist the flow of the dialogue
L256 : Please check the accuracy of the comma usage here. I experienced difficulty in understanding the number and names of the biome types
L258 : Please add “a” before “large number”
L265 : Please remove the “the” before “end product inhibition”
L268 : Consider removing “enormous”
L278 : Please add a comma after “i.e.”
L280 : Please replace “where” with “on which”
L285 : Please replace “where” with “in which”
L289 : Consider replacing “It” with an appropriate noun for clarity such as “This page”
L292-293 : Please replace “all the enzymes of the database, or enzymes” with “all enzymes in the database, enzymes”
L295 : Please add a “the” before “search”
L296 : Consider replacing “namely,” with a colon
L297 : Please fix the backwards singe quote before “HMM E-value”
L307 : Please remove the extra space before “Primary”
L313 : Please replace “the first step” with “a first step”
L314 : Please remove “have”
L331 : Please replace “point towards” with “indicates”; consider removing “enormous”
L334 : Please remove “existing”
L337 : Please correct the spelling of “wode” to “wide”
L340 : Please replace “help the” with “assist”
L353 : Please clarify the first sentence. The current structure reads as if the funding performed the work
L372 and 374 : Species and genus names should be italicized. When the entire title of the paper is italicized, please do not italicize the species and genus names
L390 : Please check the formatting of this reference

---

## Round 0.2 · accepted · Accept

I can confirm that the authors have made the modifications indicated by the reviewers. I believe the manuscript improved and can be accepted for publication as it is.